# Ultrasound Measurement of Tumor-Free Distance from the Serosal Surface as the Alternative to Measuring the Depth of Myometrial Invasion in Predicting Lymph Node Metastases in Endometrial Cancer

**DOI:** 10.3390/diagnostics11081472

**Published:** 2021-08-14

**Authors:** Marcin Liro, Marcin Śniadecki, Ewa Wycinka, Szymon Wojtylak, Michał Brzeziński, Agata Stańczak, Dariusz Wydra

**Affiliations:** 1Department of Obstetrics and Gynecology, Gynecologic Oncology and Gynecologic Endocrinology, Medical University of Gdańsk, 80-210 Gdańsk, Poland; mliro@gumed.edu.pl (M.L.); stanczak.agata.med@gmail.com (A.S.); dgwydra@gumed.edu.pl (D.W.); 2Department of Statistics, Faculty of Management, Gdańsk University, 81-824 Sopot, Poland; ewa.wycinka@ug.edu.pl; 3Department of Pathology, Medical University of Gdańsk, 80-214 Gdańsk, Poland; swojtylak@gumed.edu.pl; 4Department of Gynecologic Oncology, PCK Marine Hospital in Gdynia, 81-519 Gdynia, Poland; m.brzezinski@gumed.edu.pl

**Keywords:** ultrasound, endometrial cancer, lymph nodes metastasis, myometrial invasion, tumor-free distance

## Abstract

Background: Ultrasonography’s usefulness in endometrial cancer (EC) diagnosis consists in its roles in staging and prediction of metastasis. Ultrasound-measured tumor-free distance from the tumor to the uterine serosa (uTFD) is a promising marker for these diagnostic and prognostic variables. The aim of the study was to determine the usefulness of this biomarker in locoregional staging, and thus in the prediction of lymph node metastasis (LNM). Methods: We conducted a single-institutional, prospective study on 116 consecutive patients with EC who underwent 2D transvaginal ultrasound examination. The uTFD marker was compared with the depth of ultrasound-measured myometrial invasion (uMI). Univariable and multivariable logit models were evaluated to assess the predictive power of the uTFD and uMI in regard to LNM. The reference standard was a final histopathology result. Survival was assessed by the Kaplan–Meier method. Results: LNM was found in 17% of the patients (20/116). In the univariable analysis, uMI and uTFD were significant predictors of LNM. The accuracy was 70.7%, and the NPV was 92.68% (OR 4.746, 95% CI 1.710–13.174) for uMI (*p* = 0.002), and they were 63.8% and 89.02% (OR 0.842, 95% CI 0.736–0.963), respectively, for uTFD (*p* = 0.01). The cutoff value for uTFD in the prediction of LNM was 5.2 mm. The association between absence of LNM and biomarker values of uMI < 1/2 and uTFD ≥ 5.2 mm was greater than that between the presence of metastases and uMI > 1/2 and uTFD values <5.2 mm. In the multivariable analysis, the accuracy of the uMI–uTFD model was 74%, and its NPV was 90.24% (*p* = non-significant). Neither uMI nor uTFD were surrogates for overall and recurrence-free survivals in endometrial cancer. Conclusions: Both uMI and uTFD, either alone or in combination, were valuable tools for gaining additional preoperative information on expected lymph node status. Negative lymph nodes status was better described by ultrasound biomarkers than a positive status. It was easier to use the uTFD rather than the uMI measurement as a biomarker of EC invasion, and the former still maintained a similar predictive value for lymph node metastases to the latter at diagnosis.

## 1. Introduction

Endometrial cancer (EC) is the most prevalent malignancy of the female genital tract among the top-ranked countries on the Human Development Index (HDI) and is the tenth most common malignant neoplasm worldwide according to Global Cancer Statistics [1,2]. EC is traditionally divided into two pathogenetic types, determined by the histological and molecular features of the tumor, one, with a low risk of lymph node metastases (LNM) (type I) and the other with a high risk of LNM (type II) [3]. In 2013, the European Society of Medical Oncology (ESMO) introduced three LNM risk groups: low, intermediate, and high [4]. More recently, a conference between the European Society of Medical Oncology (ESMO), the European Society of Radiotherapy and Oncology (ESTRO), and the European Society of Gynecologic Oncology (ESGO) established a consensus that there are five LNM risk groups: low, intermediate, high-intermediate, high, and advanced metastatic, highlighting the role of ultrasonography in the assessment of the biological behavior of tumors [5]. Most of the same bodies (including the European Society of Pathology, ESP) also identified the same five prognostic groups but, based on molecular classifications, recommended expert transvaginal ultrasound examination as a mandatory element in the preoperative work-up and pointed to the greater value of this examination when conducted by an experienced gynecologist than magnetic resonance imaging [6]. Thus, ultrasonography, the most extensively used diagnostic method worldwide, meets one of the most common cancers in the world. Underscoring the importance of this method is the creation of a dictionary of changes in the endometrium and uterine cavity, created by the International Endometrial Tumor Analysis (IETA) and Morphological Uterus Sonographic Assessment (MUSA) groups [7,8]. The diagnostic capabilities of ultrasound in uterine neoplasms start from a wider range of research usefulness—including the definition of tumor types, the lexicon of uterine changes, and the concept of the Uterus Imaging Reporting and Data System (UI-RADS) [8,9,10,11]. In the late 1990s, it was noted that the depth of myometrial invasion (DOI), when measured quantitatively as a distance from the endo-myometrial junction to the deepest point of myometrial invasion, is an important predictive factor of LNM [9]. This thesis further evolved after reporting showed tumor-free distance (TFD), a measure of free muscle (as distance from the deepest point of myometrial invasion to the nearest serosal surface), to be more efficient than DOI in predicting recurrence and, thus, enabling better estimations of the actual LNM risk [10,12]. However, the opposite results were also noticed [13,14,15,16]. The difference between DOI and pTFD is shown in Appendix A. The parameter most frequently used by pathologists and gynecological surgeons is myometrial invasion—pathologically measured (pMI) and ultrasound measured (uMI). The qualitative definition of the invasion depth in relation to the full myometrial thickness is expressed as an invasion of the tumor greater than or equal to (expressed as ≥50% or ≥½) or less than (expressed as <50% or <½) the thickness of the uterine wall. Scientific work on nodal metastases, including molecular analysis, has shown the need to redefine the risk groups for endometrial cancer metastasis, especially because the “low-risk” groups under the previous criteria, recorded a higher-than-expected percentage of lymph node metastases. Due to the need to calculate and possibly reduce areas of uncertainty, some studies have taken the form of predictive model analyses, and the results evolved into a “personomics” of endometrial cancer [17,18,19,20,21].

Against the above background, the question that arises is what role do ultrasound biomarkers play in the new endometrial cancer risk groupings? Do they still have value or is their value exhausted? Is there also the possibility of redefining ultrasound markers to establish a new role for them? We wish to answer these questions by analyzing two key biomarkers used in the local staging of endometrial cancer: uMI and uTFD.

## 2. Materials and Methods

An abstract of this study with a preliminary report on 86 of the patients was published in the 24th World Congress on Ultrasound in Obstetrics and Gynecology publication of abstracts [22]; however, the current full article reporting on 116 patients with follow-up has not been previously published. The study was approved by the Research Ethics Committee of the Medical University of Gdansk (No. NKBBN/121/2014). Each subject of the study voluntarily gave their written informed consent prior to participation in the study. The study was conducted in accordance with the ethical guidelines of the Declaration of Helsinki [23]. The Standards for Reporting Diagnostic Accuracy (STARD) guidelines were followed in reporting our study results [24]. The authors declare no conflict of interest.

### 2.1. Study Design and Participants

This was a prospective clinical study comprising one hundred and sixteen consecutive patients hospitalized in the Department of Gynecology, Gynecologic Oncology, and Gynecologic Endocrinology at the Medical University of Gdansk from January 2011 to November 2012. Patients with myoma or adenomyosis were excluded from the study before commencing analysis. The study participants were either referred from outpatient health care units, or other hospitals, or were already in our cancer care registry (diagnosed in our outpatient clinic). The patients (mean age 63 ± 8.4 years) had histologically confirmed endometrial carcinoma, either by dilation and curettage (D&C) or by hysteroscopy prior to surgery. The cancer stages at diagnosis were I–III according to the International Federation of Gynecology and Obstetrics (FIGO) classifications (2009) [25]. Surgical treatment was performed in accordance with Mayo Clinic (Rochester, MA, USA) algorithms for EC [16,26]. Additional to the algorithm, sentinel lymph node biopsy (SLNB) was used in the “low-risk” group of patients (as a double method of identification; see the description below), with the addition of the SLNB procedure to full LND in the “high-risk” group of cases. In order to update the results of the study, patients were divided into five groups according to the ESMO–ESTRO–ESGO guidelines but with the proviso that molecular tests were not performed, and therefore those results are unknown [6]. 

### 2.2. Ultrasound Examination

Each patient underwent 2D transvaginal ultrasound examination, performed by a sonologist experienced in gynecologic oncology, using one Philips HD7 device (Koninklijke Philips N.V., Eindhoven, The Netherlands) with a vaginal probe (6–10 MHz). Each patient was analyzed in relation to two ultrasound (u) markers: myometrial invasion (uMI) and tumor-free distance (uTFD). Ultrasound MI was measured by subtracting the thickness of the tumor (i.e., perpendicular to the long axis) from the endo-myometrial junction to the serosa. Measurement of the uTFD was performed at the most locally advanced portion of the tumor (the tumor front) in the same three planes, taking into consideration the minimum distance reached into the serosa. Figure 1a–c shows how the uMI and uTFD measurements were performed and specifically where the measurements were taken. All measurements were carried out using a tension-free technique to avoid tissue shrinkage. The initial histopathological result was known by the sonologist. 

### 2.3. Surgical Procedures

The types of surgery undertaken were as follows: (1) simple hysterectomy and bilateral salpingo-oophorectomy with sentinel lymph node biopsy (SLNB), or, alternatively, sampling of regional LNs (in cases of unsuccessful SLNB procedures), in grades 1 and 2 endometrioid-type tumors and in cases where ultrasound assessment indicated uMI < 50%; (2) type C hysterectomy with salpingo-oophorectomy and total pelvic as well as para-aortic lymphadenectomy (LND) in patients with known risk factors of recurrence and an elevated risk of LNM (including guidance by SLN concept): serous EC, grade 3 endometrioid subtype, uMI ≥ 50%, cervical involvement. In patients with contraindications for more extensive LN surgery (e.g., poor general condition, comorbidities, morbid obesity, or advanced age) escalation of lymphadenectomy was abandoned (and SLNB was performed).

### 2.4. Sentinel Lymph Node Identification

The SLN concept was based on the combined method: (1) Tc99m-nanocolloid (1 mL/patient, activity of 18.5 MBq, Nano-Albumon, Medi-Radiopharma, Érd, Hungary) was administered into the cervical submucosa approximately 10–15 min prior to skin incision; and 2) blue dye (4 mL/patient, Oterop Methylenblau 1 mg/1 mL, Sterop Pharmacobel, Anderlecht, Belgium) was administered into the subserosa of the uterine fundus intraoperatively. During surgery, the color of the nodes and the radiotracer uptake were assessed—SLNs were defined as those that were dyed blue and/or those that had an uptake 10-fold greater than the background (handheld device, Neoprobe 2000, Neoprobe Corporation, Cincinnati, OH, USA).

### 2.5. Specimens and Samples

The pathologist was blinded to the results obtained by the ultrasonographer and was only aware of the preoperative pathologic diagnosis. All findings from external units were validated internally by our institution’s pathologist. In cases where we received material from outside sources, we asked the external organization to send those histology blocks and/or slides (in cases where there were no tissue blocks) that had undergone pathological processing and verification by the originating organization’s pathologist. Subsequently, all resected lymph nodes underwent routine histopathology processing (reference standard). Staging was defined postoperatively in accordance with FIGO (2009) classification criteria [25].

### 2.6. Statistical Analysis

Univariable logit models were evaluated for both ultrasound and histological measurements. Two quantitative predictors (uTFD and pTFD) and four qualitative predictors (uMI, pMI, grading, and cancer histology) were used. The multivariable model was built with ultrasound parameters only. The discrimination ability of models was assessed using the receiver operating characteristic curve (ROC curve) and the area under the curve (AUC). Accuracies were calculated for points of predictors that maximized the Youden index. The likelihood ratio test (LRT) was used as a global test for the models. Kaplan–Meier estimator was used to estimate the overall survival and recurrence-free survival of patients from the date of surgery to the date of death or the date of first recurrence, or the last observation date (observation censored). Calculations were made in Statistica v. 13 software.

## 3. Results

The detailed characteristics of the study population and measurements performed are presented in Table 1. The highest percentage of cases (89%) was early stage EC (confined to the uterine corpus). The most frequent histological type was endometrioid adenocarcinoma. Eighty-six patients had G1 or G2 tumors (74%). 

Table 2 shows the parameters studied in the univariable analysis, and Table 3 shows the univariable and multivariable logit models of ultrasound parameters studied with corresponding values of accuracy (ACC), area under the curve (AUC) with 95% confidence intervals (CI), sensitivity, specificity, negative (NPV) and positive (PPV) prognostic values, and p values of given parameter.

Figure 2 shows multiple correspondence analysis of ultrasound biomarkers and lymph nodes metastasis. The association between absence of LNM and biomarker values of uMI < 50% and uTFD ≥ 5.2 mm was greater than that between the presence of metastases and uMI ≥ 50% and uTFD values <5.2 mm. Appendix A shows the multivariable analysis of ultrasound factors in the low-risk group. Appendix A shows the agreement between uMI and pMI. In all, 76.6% of MI cases measured by ultrasound were consistent with MI measured by histopathology. Of these, 64% were correctly defined for MI ≥ 50%, while 84.5% were correctly defined for MI < 50%. Appendix A shows the concordance between uTFD and pTFD with use of intraclass coefficient correlation and Bland–Altman plot (ICC = 0.676, 95% CI (0.564–0.764).

Post hoc analysis (Appendix A) shows that differences between all the groups are significant: with the increase of risk, the mean rank value of uTFD decreases. In Figure 3b, the median values are shown.

### Survival Analysis and Ultrasound Biomarkers

Comparing the survival curves for uMI (less than, or equal to, or greater than 1/2 or 50%) and uTFD (less than, or equal to, or greater than 5.2 mm) cases showed no statistically significant differences (Appendix A). Appendix A shows complete and relapse-free survival for the groups with and without lymphatic metastases. The differences in survival are statistically significant for overall survival, but they are of borderline significance for relapse-free survival.

## 4. Discussion

The aim of the study was to test the predictive power of ultrasound biomarkers (uMI, uTFD) of uterine infiltration currently used to determine the risk of lymph node metastasis in endometrial cancer, taking into consideration endometrial cancer risk groupings and recurrence-free and overall survival.

In our study, lymph node involvement was found in 20 of 116 (17%) patients. In the univariable analysis, both uMI and uTFD were found to be significant predictors of LNM.

In the multivariable analysis, the model fit proved better than in the univariable analysis. The fact that the parameters were insignificant in the multiple model was the result of the strong correlation of the predictors. Thus, choosing between the uTFD or uMI ultrasound markers can be arbitrary. However, it should be noted that the above-mentioned ultrasound predictors are of greater value for indicating the absence of metastases, i.e., with uMI < 50% and/or uTFD ≥ 5.2 mm, it can be said that there will be no metastasis with a higher level of confidence than indicated with uMI ≥ 50% and/or uTFD < 5.2 mm.

There were no patients with LNM = 1 in the low-risk group, and the models were not better suited for the intermediate-risk group than for the entire sample. The higher the risk, the lower the mean uTFD values; however, in the intermediate- and high-risk groups the cutoff point for the uTFD was unchanged at 5.2 mm (in the low-risk group it could not be determined because there were no cases of LNM = 1). The frequency of uMI ≥ 50% increased in correlation with the increased risk levels, but the prediction based on uMI was better for the whole sample of patients than for the intermediate group only (with the high-risk group there were too few cases to apply the model).

The differences in the recurrence-free survival and overall survival data were not statistically significant between patients under and over the uMI ≥ 50% threshold and cutoff value for uTFD.

Our study came with limitations. Firstly, most patients did not have systematic LND, which may produce bias. As a result, we had a homogeneous group of patients for whom we possessed ultrasonography data but incomplete data on lymph nodes. However, we followed the guidelines for “low-risk” cancer cases by proceeding with de-escalating lymph node surgery [5,6,16,26]. Ultrasound examination is comparable with (or better than) MRI and sufficient for staging, especially for non-extra-pelvic disease [5,6,27]. In stage I, “low-risk” tumors (G1, G2), systematic lymphadenectomy is not recommended. For the “intermediate-risk” tumors (G3 irrespective of MI), lymph node staging is suggested, and SLNB is an option. Systematic lymphadenectomy is recommended for “high-risk” tumors (G3 and MI > 50% non-endometrioid type) [5]. Secondly, we did not include adenomyosis and myoma patients due to the increased risk of inaccurate assessment of infiltration [28]. Appendix A show ultrasound and MRI images of the uterus with endometrial cancer in adenomyosis and uterine myoma, respectively. Thirdly, the significance of ultrasound biomarkers for recurrence rates and overall survival requires further evaluation with a larger study population, including a greater number of complete observations.

In our study, to maintain correct clinical study procedures, we chose not to incorporate histological data in the analysis of two ultrasound biomarkers (uMI and uTFD). The studies we cited above recommend clinically measured (u)MI as a guide for decisions on whether to perform lymphadenectomy. Myometrial invasion (uMI) is an approved indicator for decisions on the scope of the operation in cases of potentially high-risk metastatic tumors. The limited compatibility observed in the literature between ultrasound and histopathologic assessment of tumor invasion may result from numerous factors [29,30,31,32,33]. Single ultrasound parameters have been suggested as markers for preoperative predictions of the actual extent of tumor invasion. For instance, this was achieved by employing the arcuate vascular plexus positioning or peak systolic velocity that correlate with pathologically measured depth of invasion [34,35]. Nevertheless, several ultrasound-based variables were identified as responsible for “staging errors” in EC [28,36]. Two of these included an interaction between the tumor and the myometrium: tumor size and myometrial invasion (MI). In our study, uMI but not uTFD was the interaction parameter. In contrast, parameters with higher degrees of complexity are based on a combination of different assessment methods and mathematical components. De Smet et al. [37] developed logistic regression and least squares support vector machine models with linear and radial basis function kernels built on TVUS-based factors, i.e., depth of invasion (uDOI) [38]. More recently, a two-step strategy was proposed for identifying patients at higher LNM risk. It is based on preoperative grade and two logistic regression models created on selected “objective” ultrasound variables and also includes subjective assessment of uMI and ultrasound-measured cervical stromal invasion on TVUS [39]. The models of this two-step strategy were only compared with the preoperative grading using earlier models proposed by De Smet [37] and Karlsson [40]. It was revealed that preoperative grading may omit substantial numbers (64%) of actual “high-risk” ECC cases, whereas using both models is shown to limit omissions to 17–22% of high-risk cases. Valentin has pointed out the increasingly prominent diagnostic role played by transvaginal ultrasonography in distinguishing low- and high-risk ECCs, compared with other diagnostic modalities, because it is less costly, readily available, and less time consuming [41]. It seems that the only major problem that remains is to identify uMI ≥ 50% correctly and repeatedly, because this measurement is highly variable [28,29,30,31,32,33]. Moreover, in some LNM risk estimation models, the uMI cutoff value is defined as less than 1/3 or greater than 1/3 [20]. On the other hand, a study by Antonsen et al. failed to show results of higher diagnostic value by magnetic resonance imaging (MRI) or positron emission tomography (PET)/computed tomography (CT) scanning; and the authors suggested that it is impossible to directly detect pelvic lymph nodal metastases in TVUS [42]. Studies solely based on modalities such as MRI or PET/CT, are likely to determine EC metastatic status [42,43,44]. The authors of the previously cited study also suggested that MRI should be included in the routine preoperative work-up of patients with EC [43]. Some researchers point to a particularly high PPV (93.3%) in PET/CT for LNM assessment in “high-risk” tumors [44].

The best correlation between ultrasound and pathology should be expected in the “expanding type” of tumor growth. This type is characterized by a broad front of neoplastic infiltration with a sharp demarcation of tumor tissue from the adjacent healthy tissues. This margin should be clearly identified by invasion markers such as uMI, uTFD, or endo-myometrial irregularity. Among these markers, the latter seems to be the most subjective and difficult to assess. In physiology, the endo-myometrial junction is involved in facilitating sperm transport through the modulation of uterine peristalsis and the implantation of the blastocyst, thus it influences fertility [45]. However, its role in oncology is not yet well elucidated. This intermediate zone is lost during EC invasion. Therefore, it is possible that the structure may be a helpful indicator of early EC invasion. Measurement of endo-myometrial irregularity was included in the “REC” (risk of endometrial cancer) scoring system by Dueholm et al. [46], which indicates malignancy in cases of postmenopausal bleeding and endometrial thickness ≥5 mm. Molecular studies seem to confirm the potential role of this zone in the invasion of cancer in which process the HOX genes may be involved [47,48]. Tumor volume was included in scoring systems by Mitamura et al. and Imai et al., who further developed the results of earlier studies by Todo et al. [49,50,51], although tumor volume was measured by MRI, and their scoring system was a mixture of clinical and pathological features. Tumor infiltration should be treated as a 3D structure, and what is initially visible during examination may be inconsistent with the image when altering the examination plane (since ultrasound examination is one dimensional, and microscopic examination is multidimensional). That is why, uMI, uTFD, and other one-dimensional and/or single-plane parameters may be more observer-dependent than tumor volume [52]. Moreover, the localization of tumors may cause discrepancies in the proper evaluation of the ultrasound parameters [29,36]. The issues stated above mainly refer to type I EC. In serous carcinomas, the deepest point of neoplastic infiltration is often synonymous with the deepest localized embolus in the lymphatics [15]. Many serous ECs present an image of only polypoidal growth, accompanied by broad peritoneal metastasis. However, the most controversial cases are endometrioid G3 tumors, which belong to type I EC, but may histologically represent a heterogenic group of cancers with frequent multiplication of biological features typical of type II EC [49,53].

uTFD is more feasible than uMI in assessing locoregional EC invasion and achieves a similar level of accuracy. The additional measurement of uTFD may be recommended, especially when the measurement of uMI alone is difficult. Exploring new models that include other ultrasound and molecular factors can improve the predictive value of preoperative research and, thus, bring us closer to biological disease staging and personalized treatments. For now, there are lymph node metastasis risk indexes that additionally assess (that is, in addition to the parameters from TVUS) such clinical features as CA-125 levels or pathological features such as LVSI, which have a higher accuracy, and therefore indicate the need to supplement TVUS with other non-imaging studies [19,20]. However, it is unfortunate that in most cases we only learn about the LVSI status after surgery, which means that the risk of recurrence is predicted more than the risk of lymph node metastases. Therefore, personomics cannot be based on clinical or ultrasound features alone [21].

The results of the survival analysis in our study with respect to lymph node status reflect its prognostic importance. This is in relation to the results of the PORTEC 2 and 3 studies, which determined the effects of adjuvant treatment, namely the use of brachytherapy and radiotherapy in the context of recurrence, depending on risk factors indicated by the histopathological examination (pMI, LVSI) [54,55]. The results of the relapse-free survival are at the borderline of statistical significance, which is probably due to the small sample size. It should be recognized that these differences would be more significant if the group were larger.

Considering the suggestions stated above, a prospective study comprising an analysis of other models with uMI and/or uTFD will be valuable. Grade did not add any value to the current analysis, which indicates that predictive ultrasound models reflect the biological activity of the tumor, for which grade is also responsible. It would be interesting to know whether the local staging of EC may be enhanced in order to indicate “high-risk” patients who may benefit from the less extensive sentinel lymph node procedure instead of systematic lymphadenectomy. As of today, it seems that both above-mentioned locoregional disease extent ultrasound biomarkers, due to their high negative predictive values, serve to emphasize the low risk of lymph node metastases. Furthermore, the high negative predictive value of these biomarkers suggests that, under the logistical constraints during the COVID-19 pandemic, ultrasound can select low-risk cases in which surgery can be delayed under the forced contingencies of the present situation, as it is in cases of breast cancer [56]. Further study is required to validate the above findings and suppositions.

## 5. Conclusions

One must distinguish between pathological and clinical prognostic and predictive models in endometrial cancer. The purpose of ultrasound is to plan appropriate treatments that avoid over-treatment and that balance benefits and risks for patients. Ultrasound allows fairly accurate assessment of disease staging to identify those patients who do not require systematic lymphadenectomy. It is easier to measure uTFD rather than uMI as a biomarker of endometrial cancer invasion, while still maintaining a similar predictive value for lymph node metastases at diagnosis. The additional measurement of uTFD may be especially recommended when the measurement of uMI alone is difficult. As the results of our study cannot be applied to patients with endometrial cancer with concomitant myomas and/or adenomyosis, it will be necessary to conduct a further study with this group of patients. There is a need for new, affordable clinical models that combine ultrasound with the assessment of molecular properties based on histopathological examination before surgery, to assess the biological stage of the disease as precisely as possible.

## Figures and Tables

**Figure 1 diagnostics-11-01472-f001:**
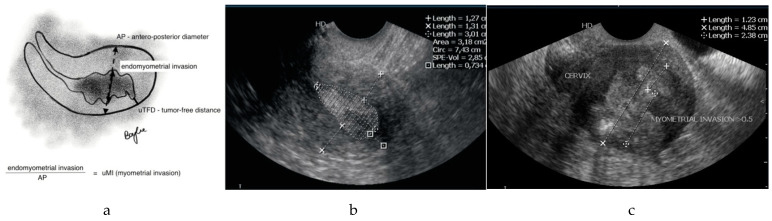
The measurements of infiltration depth (uMI) and tumor-free muscle thickness (uTFD) on transvaginal ultrasound: (**a**) graphic representation of measurements of the uMI and uTFD; (**b**) ultrasound sagittal image with the measurement of uTFD (between squares, >5.2 mm—low risk of LNM), marking the area of tumor infiltration with the measurement of the thickness of free walls; MI clearly <0.5—low risk of LNM; (**c**) ultrasound sagittal image of significant tumor invasion, MI much more than 0.5, uTFD unmeasurable (area of lowest measuring point on the photograph, the infiltration reaches the sub serosa); high risk of LNM.

**Figure 2 diagnostics-11-01472-f002:**
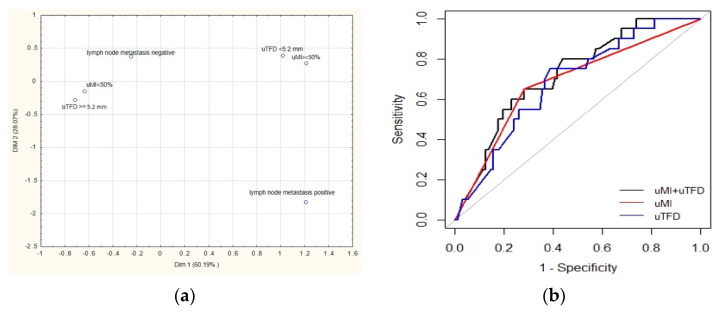
(**a**) Multiple correspondence analysis of ultrasound biomarker status and lymph node. (**b**) Presents the associations between ultrasound parameters and risk groups (ESGO 2012, due to the existence of this risk stratification during the study). There was a statistically significant association between uMI and uTFD and risk groups (Figure 3a,b).

**Figure 3 diagnostics-11-01472-f003:**
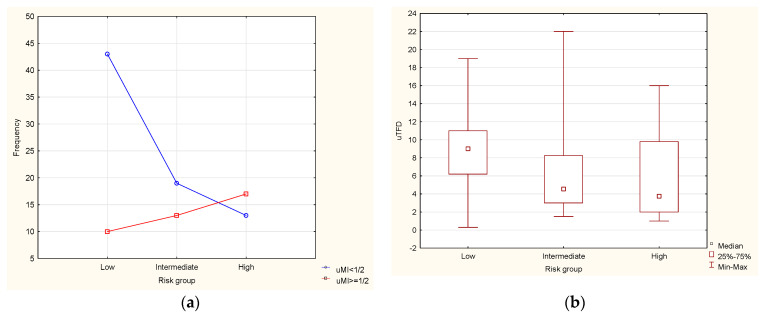
Frequency of patients in risk groups according to uMI (*p* = 0.00172 for chi-square test of independence) (**a**), and box-plots of distributions of uTFD for risk groups (Kruskall–Wallis ANOVA *p* = 0.0006) (**b**).

**Table 1 diagnostics-11-01472-t001:** Baseline characteristics of the study population (116 patients).

Variable	Characteristic	Value
**Age at diagnosis (range)**	Mean ± SD (range)	63 ± 8.3 (40–85)
FIGO stage *	Number (%)	
Ia		69 (59)
Ib		35 (30)
II		5 (5)
III		7 (6)
**Histologic type**	Number (%)	
Endometrioid		82 (71)
Endometrioid with epithelial differentiation		20 (17)
Serous carcinoma		11 (9)
Carcinosarcoma		3 (3)
**Grade**	Number (%)	
1		41 (36)
2		45 (39)
3		28 (25)
**Risk grouping according to initial risk ***	Number (%)	Number of patients with metastatic nodes (%)
Low	54 (46)	1 (2)
Intermediate	32 (27.5)	7 (22)
High	30 (26)	12 (40)
**uTFD [mm]**	Mean ± SD (range)	7.39 ± 4.83 (0.3–22.0)
**uMI**	Number (%)	
<50%		76 (66)
≥50%		40 (34)
**Lymph node procedure**	Number (%)	
SLNB only		70 (60)
LND (+SLNB)		46 (40)
**Lymph nodes extracted**	Number	1298
SLNB cases	Number (%)	313 (24)
LND (+SLNB) cases	Number (%)	985 (76)
**Lymph nodes metastases**	Number of patients (%)	20 (17)
**Distribution of nodes:**	Number (%)	34/1288 (2.64)
Obturator	19 (7 SLN)	
Iliac nodes	13 (2 SLN)	
Para-aortic	2	
**Risk grouping according to ESGO–ESTRO–ESP guidelines ***	Number (%)	Number of patients with metastatic nodes (%)
Low	52 (45)	0 (0)
Intermediate	30 (26)	4 (3)
High-intermediate	21 (18)	4 (3)
High	13 (11)	12 (10)
Advanced metastatic	0 (0)	0 (0)

ESGO—European Society of Gynecological Oncology, ESP—European Society of Pathology, ESTRO—European Society of Radiotherapy and Oncology, FIGO—International Federation of Gynecology and Obstetrics, LND—lymphadenectomy, uMI—ultrasound measured myometrial invasion, SLNB—sentinel lymph node biopsy, uTFD—ultrasound-measured tumor-free distance, TVUS—transvaginal ultrasound; * FIGO stage refers to FIGO staging 2009–2018; * no data available.

**Table 2 diagnostics-11-01472-t002:** Univariable analysis of the value of predictive factors for lymph node metastasis in the study group.

Variable		OR (95% CI)	*p* Value	Significance (α = 0.05)	ACC	Specificity	Sensitivity	NPV	PPV	AUC
**Ultrasound parameter**	(u)MI (≥50%)	4.746 (1.710–13.174)	0.0028	Yes	70.7%	71.88%	65.0%	92.68%	22.67%	0.684 (0.568–0.801)
	(u)TFD	0.842 (0.736–0.963)	0.0119	Yes	63.8%	61.46%	75.0%	89.02%	35.48%	0.683 (0.563–0.803)
**Histologic parameter**	(p)MI (≥50%)	6.600 (2.196–19.833)	0.0008	Yes	69.83%	68.75%	75.0%	92.18%	28.84%	0.719 (0.611–0.827)
	(p)TFD	0.843 (0.747–0.950)	0.0052	Yes	77.59%	82.29%	55.0%	90.79%	32.50%	0.712 (0.577–0.846)
**Grading**	G1	1			47.41%	39.58%	85.0%	89.77%	39.29%	
	G2	2.667 (0.657–10.825)	0.1700	No						0.673 (0.550–0.791)
	G3	5.700 (1.386–23.449)	0.0159	Yes						
**Cancer histology**	Endometroid	1			74.34%	78.49%	55.0%	92.96%	33.34%	
	Endometroid with squamous differentiation	2.704 (0.793–9.216)	0.111	No						0.685 (0.559–0.811)
	Serous	9.733 (2.463–38.459)	0.001	Yes						

ACC—accuracy (according to the highest Youden index), AUC—area under the (ROC) curve, ESGO—European Society of Gynecological Oncology, ESP—European Society of Pathology, ESMO—European Society of Medical Oncology, ESTRO—European Society of Radiotherapy and Oncology, G—grade, MI—myometrial invasion, NPV—negative predictive value, PPV—positive predictive value, TFD—tumor-free distance, u—ultrasound (measure), p—pathomorphological (measure).

**Table 3 diagnostics-11-01472-t003:** Multivariable analysis of the value of ultrasound predictive factors for lymph node metastasis.

Variable		OR (95% CI)	*p* Value	Significance (α = 0.05)	ACC	Specificity	Sensitivity	NPV	PPV	AUC
**Multivariable analysis**										
**Ultrasound parameter**	(u)MI (≥50%)	0.470 (0.060–3.667)	0.471	No	74.13%	77.08%	60%	90.24%	35.29%	0.722 (0.607–0.836)
	(u)TFD	0.950 (0.745–1.23)	0.683	No						
	(u)MI × (u)TFD	0.938 (0.683–1.287)	0.691	No						

ACC—accuracy (according to the highest Youden index), AUC—area under the (ROC) curve, ESGO—European Society of Gynecological Oncology, ESP—European Society of Pathology, ESMO—European Society of Medical Oncology, ESTRO—European Society of Radiotherapy and Oncology, G—grade, MI—myometrial invasion, NPV—negative predictive value, PPV—positive predictive value, TFD—tumor-free distance, u—ultrasound (measure), p—pathomorphological (measure).

## Data Availability

The collected clinical material is in electronic form in the clinic’s repository and may be made available in an anonymized form upon request.

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
