# Peer review of "Ultrasound Measurement of Tumor-Free Distance from the Serosal Surface as the Alternative to Measuring the Depth of Myometrial Invasion in Predicting Lymph Node Metastases in Endometrial Cancer"

_diagnostics, 2021, doi:10.3390/diagnostics11081472_

Round 1

Reviewer 1 Report

This paper determines the usefulness of Ultrasound-measured tumor-free distance from the  tumor to the uterine serosa (uTFD) in locoregional staging, and thus in the prediction of lymph node metastasis (LNM)

Strengths:

The sample size and the study design and participants.

Weaknesses:

This work should focus more on the objective of evaluating uTFD rather than adding other parameters that mislead the reader's interest, such as survival analysis.

The lack of inclusion of patients with fibroids or adenomyosis may modify the results. This could mean that the results obtained in this study were not applicable since many patients with endometrial cancer have fibroids and / or adenomyosis.

The authors should relate uTFD with pTFD, to check the concordance, for this, intraclass coefficient correlation or Bland-Altman plot could be used.

They should estimate the predictive value of uTFD separately, MI separately and the combination of both (they have it, but they lack the interaction of the two variables in the multivariable model).

The results should be presented with graphs. They should add a plot with the AUC with three lines, one for uMI alone as a predictor of LNM, another with uTFD alone as a predictor of LNM, and another with the AUC of the multivariate model with the two variables, to assess how much each parameter contributes separately and how much the two together.

Throughout the article you should put multivariable not multivariate.

In the tables it is not necessary to put p-value, and significance, with the confidence interval of the OR it is already enough, but it should add the AUC of the ROC curve.

Figure 1 is not illustrative of a sagittal image. The uterine cavity is not visible in its entirety. The graphic representation is in a coronal cut, while the ultrasound image is in a poorly made sagittal. Where should the measures be taken?

Line 189: Delete 74%, write 75%

Reviewer 2 Report

I recommend a minor revision and a better synthesis of the results.

Round 2

Reviewer 1 Report

Dear authors,

We believe that with the changes made the article has improved. Thanks for the job.

Reviewer 2 Report

A summary presentation is preferable in the future!